# New Insight into the Mechanism for the Removal of Methylene Blue by Hydrotalcite-Supported Nanoscale Zero-Valent Iron

**Jiaqi Fan [1], Bo Zhang [1,*], Bohong Zhu [1], Weili Shen [1], Yuan Chen [2] and Fanjun Zeng [2]**

[1] College of Materials and Advanced Manufacturing, Hunan University of Technology, Zhunzhou 421000, China
[2] Linwu Shunfa Investment Group, Chenzhou 424300, China
[*] Correspondence: 13747@hut.edu.cn

**Abstract:** Nanoscale zero-valent iron (nZVI) has become a new and ecofriendly adsorbent material with promising applications. Herein, hydrotalcite-supported nanoscale zero-valent iron (nZVI@H) is synthesized for the first time and used for testing the removal of methylene blue (MB) in an aqueous solution. The successful fabrication of nZVI@H is characterized by SEM, BET, XRD, FTIR and zeta-potential analyses. The results showed that 99.6% of MB is removed using nZVI@H after 30 min of reaction at an initial MB concentration of 40 mg·L$^{-1}$, while the bare nZVI is only at 71.2%. The kinetic analysis yielded that the removal process of MB using nZVI@H is consistent with the Langmuir model and the quasi-second-order kinetic model. According to the Langmuir model, the maximum adsorption of nZVI@H on MB is 81 mg·g$^{-1}$. This study provides a new idea about the mechanism of MB removal, namely, MB is converted to the colorless LMB through an Fe$^0$ redox reaction and simultaneously attached at the surface of nZVI@H through an adsorption process, and finally removed via complexation precipitation.

**Keywords:** hydrotalcite-supported nanoscale zero-valent iron; removal; mechanism; methylene blue

## 1. Introduction

Dyes are commonly utilized in the material and chemical industries, especially in printing, dyeing, textiles and cosmetics [1]. Global annual production of dyes has exceeded 700,000 tons, of which 2~15% are discharged into the environment with industrial wastewater during the printing and dyeing process [2]. Most synthetic dyes are highly polluting, meaning environmentally durable, non-biodegradable and toxic [3]. As a result, the release of untreated dye wastewater streams into the environment can pose a considerable risk to a wide range of creatures and humans [4]. Among the many dye contaminants, the most frequently used basic cationic dye is MB, which is also a popular model for water pollutants. According to research, even minimal concentrations of MB into a body will cause serious damage including elevation of blood pressure, gastrointestinal pain, headaches and vomiting [5,6]. For the above reasons, it is particularly important to develop an efficient method for the remediation of dye wastewater.

At present, to remove dyes from wastewater, many approaches have been researched such as electrolysis [7], catalytic reduction [8], photocatalysis [9–11], membrane separation [12], chemical oxidation [13], and particularly adsorption, since it is preferred because of its simplicity of operation, low energy consumption, non-selectivity to hazardous contaminants, and high efficiency [14]. Principally, nNZVI particles are superior to other adsorbent materials because of their large specific surface area, high reducibility and activity [15,16]. Unfortunately, several technical issues remain in the application of nZVI. On the one hand, because of van der Waals forces and magnetic interactions between the particles, nZVI tends to agglomerate, resulting in a significant loss in dispersibility [17]. On the other hand, nZVI is rapidly oxidized to other substances, which reduces its activity. [18].

To address these drawbacks, the high dispersion strength of nZVI is the key to increasing its activity. A series of supported nZVI particles has been investigated, such as carbonaceous materials-supported nZVI (including biochar and activated carbon), mineral materials-supported nZVI (including kaolin and bentonite) and high polymer-supported nZVI (including polyacrylamide and sodium carboxymethyl cellulose) [19,20]. Compared to non-modified nZVI, the modified nZVI can be exhibited to perform better in treating contaminants (i.e., azo dyes and phenols) [21].

Hydrotalcite is a layered anionic clay mineral with excellent physicochemical properties. It has been widely considered as a promising dispersant due to its obvious advantages including appreciable stratified structure, large specific surface area and high mechanical strength [22,23]. More importantly, hydrotalcite weighs less than other dispersants such as activated carbon and bentonite. This feature allows for a long floating time on liquid surfaces rather than rapid precipitating. When hydrotalcite is successfully supported as a carrier, it exhibits suspension in the liquid and facilitates full contact between the adsorbent and wastewater [24]. However, there are few studies on using hydrotalcite as the carrier for nZVI and its removal of dye wastewater. Thus, the main goals of this work were to: (1) prepare and characterize analysis of nZVI@H; (2) investigate different factors affecting the adsorption of MB; (3) perform kinetic analysis; and (4) investigate the mechanism of MB adsorption.

## 2. Materials and Methods

### 2.1. Materials and Chemicals

Sodium borohydride (>99%), ferric chloride hexahydrate (>97%), methylene blue (>99%), and ethanol (>99.5%) were purchased from Tianjin Fuchen Chemical Company (Tianjin, China) and hydrotalcite ($Al_2Mg_6(OH)_{16}CO_3 \cdot 4H_2O$) was purchased from Shandong Youso Chemical Technology Co. Ltd., (Linyi, China) with the purity of 99.6%. All reagents used in the studies were of analytical quality, and the water used was deionized water.

### 2.2. Synthesis of nZVI@H

The liquid-phase reduction process was used to prepare the nZVI, and nZVI@H was synthesized by adding hydrotalcite as a support material with synthesis of different nZVI@H by varying mass ratios of hydrotalcite and iron ($M_{hydrotalcite}:M_{iron}$ = 1:2, 1:1 or 2:1). Briefly, in a 500 mL three-neck flask, a 100 mL mixture of ethanol and deionized water (80%, *v/v*) was added to dissolve 1.21 g of $FeCl_3 \cdot 6H_2O$, and various amounts of hydrotalcite (0.125 g, 0.25 g, or 0.50 g) were added into the solution under a mechanical stirring condition. Then, the above solution was dropwise added with a freshly prepared $NaBH_4$ solution (0.34 g of $NaBH_4$ in 50 mL) to reduce the nZVI particles. Finally, the obtained nZVI@H contained 0.25 g of nZVI. The reaction is shown in Equation (1).

$$4Fe^{3+} + 3BH_4^- + 9H_2O \rightarrow 4Fe^0(s)\downarrow + 3H_2BO_3^-(aq) + 12H^+ + 6H_2(g)\uparrow \tag{1}$$

After the addition of $NaBH_4$, the blended solution was stirred for another 30 min at room temperature without interruption. The obtained black particles were segregated using vacuum suction filtration and washed with anhydrous ethanol two times. Finally, the solid was dried at 60 °C under a vacuum condition for 48 h.

### 2.3. Materials Characterization

The morphology, particle size and specific surface areas were analyzed using scanning electron microscopy (SEM, JSM-7500F, Applied Scientific Instruments Co., Ltd. (Shanghai, China)) and a Brunauer–Emmett–Teller (BET, PM7240, Huapu Hengchuang Instruments Co., Ltd. (Beijing, China)) analyzer before and after supporting, and the composition and chemical bonding of the supported materials were identified using X-ray diffraction (XRD, D8-ADVANCE, Beijing Softron Biotechnology Co., Ltd., Beijing, China) and Fourier infrared spectroscopy (FTIR, 6600, Jiangsu Skyray Instrument Co., Ltd., Kunshan, China). In addition, zeta potential was used to detect the surface charge of the material (Malvern

Zetasizer Nano ZS90, Foshan Nanbeichao Electronic Commerce Co., Ltd., Foshan, China), and a vibrating sample magnetometer was used to measure magnetic properties (VSM, YPC7-VSM-130, Shiya Technology (Guangdong) Co., Ltd., Guangzhou, China). A UV-Vis spectrophotometer measured the absorbance of the MB solution (UV-4800, UNICO (Shanghai) Instrument Co., Ltd., Shanghai, China,) after centrifugation at high speed before and after the treatment of the removed material. The MB solution was heated using a water bath heating pot (DF-101S, Yuhua Instrument Co., Ltd., Gongyi, China).

*2.4. Batch Experiments*

Batch studies were conducted in a 500 mL flask holding 500 mL of MB solution mechanically stirred at 300 rpm at room temperature. The experiment reaction was activated when the nZVI@H was added. At regular time intervals (2 min, 5 min, 10 min, 20 min and 30 min), 10 mL of the solution was collected and centrifugated for later analysis. The main reaction conditions involved in this experiment were as follows: removal of the MB solution with a pH of 9 and a concentration of 40 mg·L$^{-1}$ at 20 °C using nZVI@H material with a mass ratio of hydrotalcite to nZVI solid (supporting ratio) of 1:1 at a dosage of 0.5 g L$^{-1}$, and a series of effect parameters, including the hydrotalcite/iron mass ratio (bare, 1:2, 1:1, and 2:1), the pH of MB (5, 7, 9, and 11), the initial concentration of MB (40 mg·L$^{-1}$, 100 mg·L$^{-1}$, and 200 mg·L$^{-1}$), the amount of adsorbent added (0.1 g·L$^{-1}$, 0.3 g·L$^{-1}$, 0.5 g·L$^{-1}$, and 0.7 g·L$^{-1}$), and the adsorbent temperature (20 °C, 40 °C, and 60 °C). The concentration of MB after centrifugation at the specified times was measured using a UV-Vis spectrophotometer.

The treated MB solution was poured into a 25 mm cuvette to measure absorbance at 664 nm using a UV-Vis spectrophotometer (the maximum UV absorption spectrophotometry of MB was 664 nm) [25]. The removal rate of MB can be calculated from Equation (2):

$$R = \frac{\rho_0 - \rho_t}{\rho_t} \times 100\% \tag{2}$$

where $R$ is the removal rate (%), $\rho_0$ is the initial mass concentration (mg·L$^{-1}$), and $\rho_t$ is the mass concentration at reaction time $t$ (mg·L$^{-1}$).

The absorbance was different for different concentrations of MB solution at 664 nm; the lower the concentration of MB, the lower the absorbance and the lighter the color of the solution, the better the removal effect. The results are shown in Figure 1, and the treatment times are 2 min, 5 min, 10 min, 20 min and 30 min from left to right. All degradation experiments were repeated thrice.

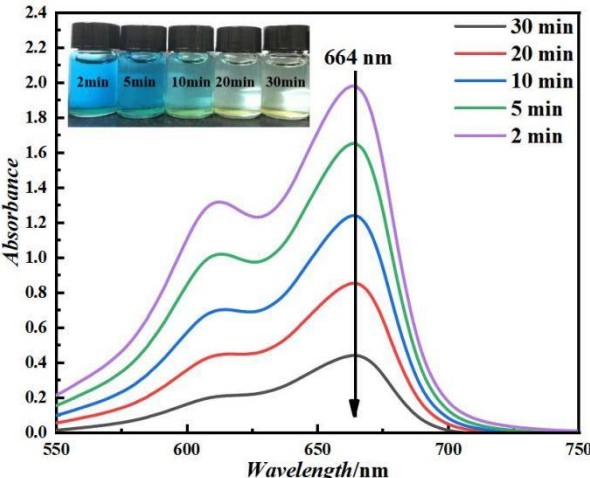

**Figure 1.** Absorbance of MB at different times.

*2.5. Isothermal Equation*

1.  The Langmuir model represents a monolayer adsorption equilibrium on the surface of a homogeneous medium and can be described in linearized form by Equation (3):

$$\frac{Ce}{Qe} = \frac{1}{bQm} + \frac{Ce}{Qm}$$

(3)

where $C_e$ is the concentration of the pollutant at equilibrium (mg·L$^{-1}$), $Q_e$ is the equilibrium adsorption of the pollutant by the material (mg·g$^{-1}$), $Q_m$ is the maximum adsorption of the pollutant by the material (mg·g$^{-1}$), and $b$ is the Langmuir adsorption constant (L·mg$^{-1}$).

2.  The Freundlich model describes multilayer adsorption processes on material interfaces. The linearized form of the Freundlich model can be described by Equation (4) [26]:

$$\log Qe = \log K_f + \frac{1}{n}\log C$$

(4)

where $K_f$, the Freundlich constant ((mg·g$^{-1}$)/(L·mg$^{-1}$)$^{1/n}$), represents the adsorption capacity of the adsorbent, $n$ is the constant (if $n > 1$, the reaction process could promote the adsorption reaction), $Q_e$ is the amount of adsorption at equilibrium (mg·L$^{-1}$), and $C_e$ is the concentration at equilibrium of the reaction (mg·L$^{-1}$).

*2.6. Adsorption Kinetics*

The experimental data for the removal of MB at different concentrations using nZVI@H were fitted using a quasi-first-order kinetics equation (see Equation (5)) and quasi-second-order kinetics equation (see Equation (6)) to study the characteristics of the adsorption process, where the quasi-first-order kinetics model represents the physical adsorption process while the quasi-second-order kinetics model describes the chemical adsorption processes. The Weber–Morris model analyzes intraparticle diffusion mechanisms (see Equation (7)).

$$\ln(q_e - q_t) = \ln q_e - k_1 t$$

(5)

$$\frac{t}{q_t} = \frac{1}{k_2 q_e^2} + \frac{1}{q_e}t$$

(6)

$$q_t = k_d t^{\frac{1}{2}} + C$$

(7)

where $q_e$ is the equilibrium adsorption (mg·g$^{-1}$), $q_t$ is the adsorption at time t (mg·g$^{-1}$), $t$ is the adsorption time (min), $k_1$ is the quasi-first-order kinetics equation adsorption rate constant (min$^{-1}$), $k_2$ is the quasi-second-order kinetics equation adsorption rate constant (g·mg$^{-1}$·min$^{-1}$), $k_d$ is the intraparticle diffusion rate (mg·(g·min$^{1/2}$)$^{-1}$), and $C$ is the intercept distance (mg·g$^{-1}$).

*2.7. nZVI@H Reusability and Stability Study*

Currently, any adsorbent has the advantages of low cost, environmental friendliness, and strong adsorption capacity; but lacking excellent reusability, it is useless. Therefore, studying the reproducible use of nZVI@H is a crucial issue. The reusable test was as follows: at the end of each adsorption cycle, nZVI@H was collected using an external magnet, washed with ethanol, and reused in the following process (MB solution parameters per cycle: C = 100 mg·L$^{-1}$, dosing amount = 0.5 g·L$^{-1}$, pH = 9, and T = 20 °C).

**3. Results and Discussion**

*3.1. Investigation of the Properties of nZVI@H*

Figure 2a shows that the bare nZVI particle is mostly spherical with the size of 90 nm. However, the nZVI particles are simple to heap and eventually form cluster-like aggregates due to van der Waals forces and magnetic interactions. These findings are congruent with

those of Arabi et al. [27]. Figure 2b shows the nZVI is clearly discrete on the layered surface of the hydrotalcite after stabilization. Thus, this suggests that the hydrotalcite is essential to the dispersal and stabilization of nZVI. Table 1 shows the BET results of hydrotalcite, nZVI and nZVI@H, respectively. As seen from the table, the nZVI particles are better dispersed due to the support of hydrotalcite, and its particle diameter decreases significantly, so its specific surface area increases from 11.6 $m^2 \cdot g^{-1}$ to 28.7 $m^2 \cdot g^{-1}$. The specific surface area of hydrotalcite decreases from 31.2 $m^2 \cdot g^{-1}$ to 28.7 $m^2 \cdot g^{-1}$ due to the attachment of nZVI particles in the lamellar structure of hydrotalcite, which block the pores.

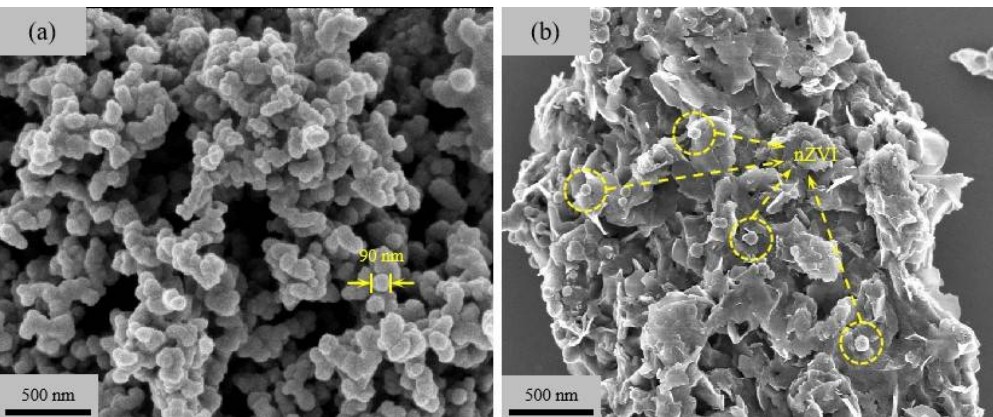

**Figure 2.** SEM images of (**a**) nZVI and (**b**) nZVI@H.

**Table 1.** The BET surface area of hydrotalcite, nZVI and nZVI@H.

| Sample | Hydrotalcite | nZVI | nZVI@H |
|---|---|---|---|
| Surface area $m^2 \cdot g^{-1}$ | 31.2 | 11.6 | 28.7 |

Figure 3 shows the XRD results of hydrotalcite, nZVI and nZVI@H, respectively. It is observed that the elements corresponding to the central characteristic peaks ($2\theta = 7.6°$, $22.8°$, $34.5°$, and $38.5°$) of the hydrotalcite spectra are C, Mg, and Al, respectively. The spectra of both nZVI and nZVI@H appear as diffraction peaks on the crystalline surface of $\alpha$-Fe (110) in the body-centered cubic structure, proving that $Fe^0$ is the main component of both, with the characteristic diffraction peak of $2\theta = 45°$~$46°$. Due to the weak crystallinity of Fe, a weaker $Fe^0$ peak is observed in nZVI@H. In the XRD spectrum of nZVI@H, the appearance of C, Mg, and Al characteristic peaks, in addition to $Fe^0$ peaks, indicates that the preparation of the composite was successful.

The Fourier spectra of the surface groups of nZVI and nZVI@H are shown in Figure 4, where nZVI and nZVI@H have peaks ranging 3400~3500 $cm^{-1}$ that are attributed to the bonding of O–H to $Fe^0$ or its oxide–hydroxide. The common spectrum at 1630~1640 $cm^{-1}$ comes from the vibration at C = O, which is caused by the adsorption of $CO_2$ in the air. The weaker peaks in nZVI and nZVI@H are 1534.21 $cm^{-1}$ and 1548.78 $cm^{-1}$, which come from the oxidation of $Fe^0$ to form $Fe_3O_4$, $Fe_2O_3$ and FeOOH on the surface, or the introduction of the anhydrous ethanol group during the preparation of the material. At the same time, a high intensity band less than 700 $cm^{-1}$ is not detected on the surface of nZVI@H compared with that of nZVI, indicating that the antioxidant properties of nZVI@H are enhanced. The enhanced oxidation resistance helps the adsorbent to remain active for a longer period of time and improves the adsorption capacity [28].

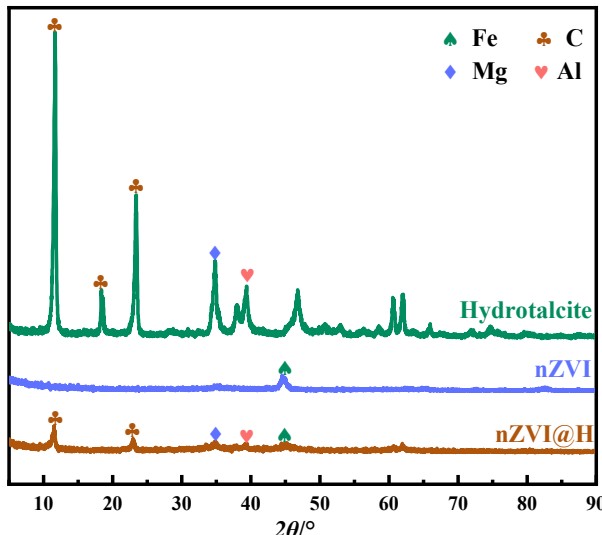

**Figure 3.** XRD images of hydrotalcite, nZVI and nZVI@H.

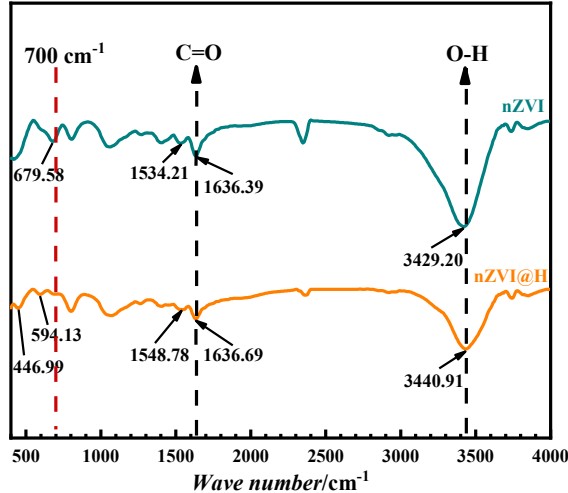

**Figure 4.** FTIR images of nZVI and nZVI@H.

Zeta-potential measurements are taken to better illustrate the role of the adsorbent in the adsorption of MB. Figure 5 shows the zero-charge point (ZCP) of nZVI@H at pH 5.7. As a result, the surface of nZVI@H will be positively charged at a pH less than 5.7 and negatively charged at a pH greater than 5.7. This reflects the fact that the negative charge on the nZVI@H surface favors the adsorption of positively charged cationic dyes such as MB at pH > 5.7. In contrast, at pH < 5.7, the surface of nZVI@H has a positive charge, causing electrostatic repulsion between the two particles of nZVI@H and MB.

### 3.2. Removal of MB

Figure 6a shows that the removal of MB using nZVI@H is significantly more effective than hydrotalcite and nZVI. The removal rates for nZVI@H reached 99.6%, while that of hydrotalcite and nZVI were 27.8% and 71.2%, respectively. This is because hydrotalcite is poorly dispersed in aqueous solutions and mostly floats on the surface of the solution. Moreover, its specific surface area is relatively small, and the adsorption sites are lower than those of nZVI, which makes it less effective, while nZVI is more prone to agglomeration and oxidation than nZVI@H, resulting in reduced activity. After supporting, the layered structure of hydrotalcite is able to trap the nZVI particles between its layers, effectively slowing down the oxidation rate and agglomeration and significantly improving the material dispersion, thus improving the efficiency of MB removal. The equilibrium adsorption

of the three materials in Figure 6b also shows that the nZVI after hydrotalcite supporting has a larger adsorption capacity and improves the removal rate of MB.

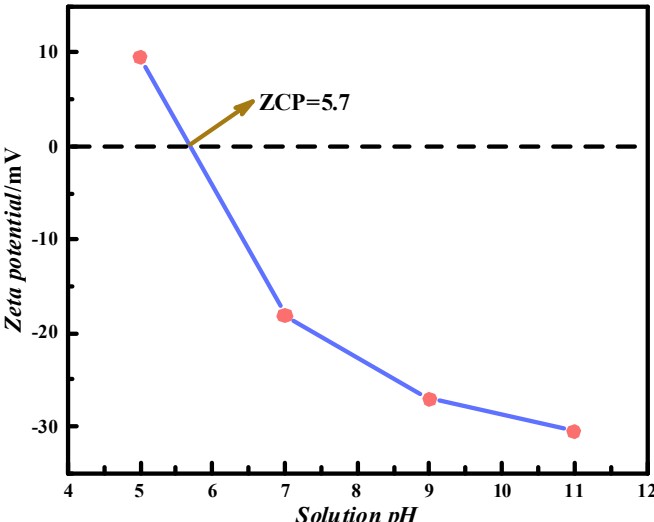

**Figure 5.** Zeta potential of nZVI@H at different pH levels.

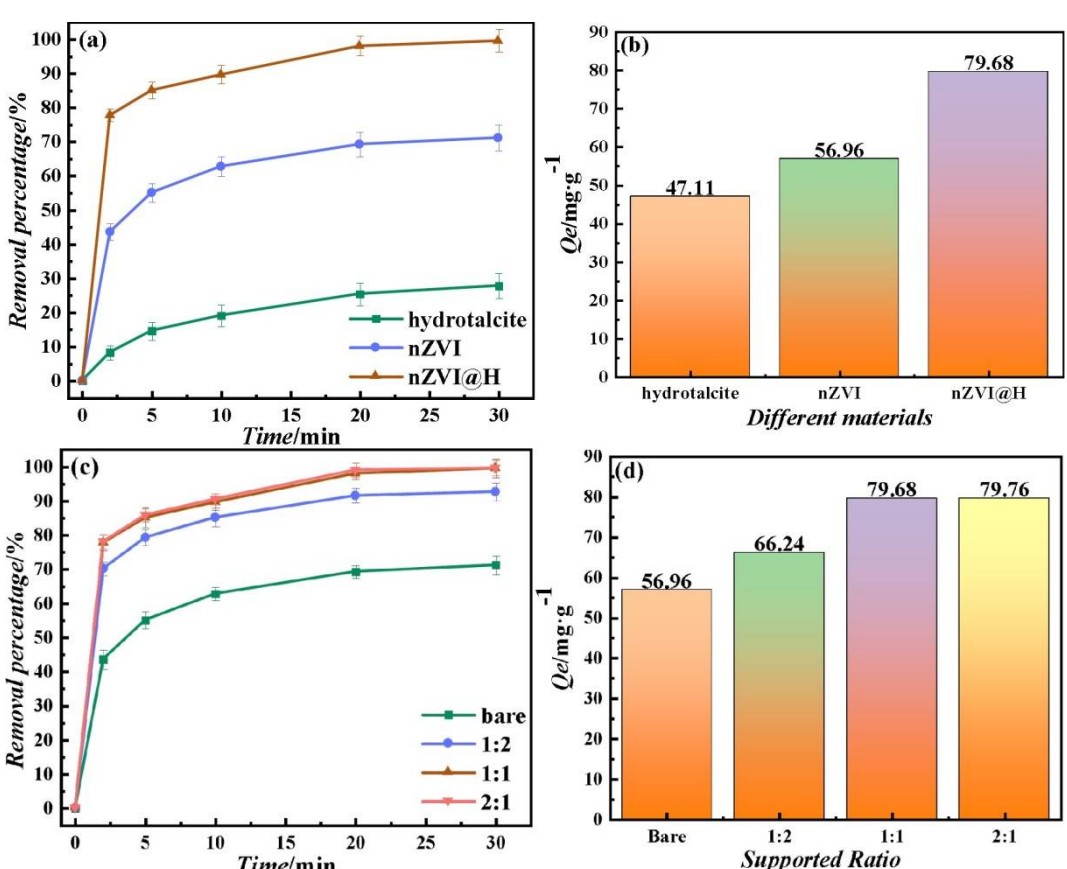

**Figure 6.** (**a**) Comparative degradation of MB using different materials. The dosage of hydrotalcite, nZVI and nZVI@H (hydrotalcite:Fe = 1:1) are 0.5g·L$^{-1}$ with a pH of 9, an initial MB concentration of 40 mg·L$^{-1}$, and a temperature of 20 °C; (**b**) equilibrium adsorption of different materials; (**c**) comparative degradation of MB using nZVI@H with various ratios (hydrotalcite/iron mass). The dosages of nZVI and nZVI@H are 0.5g·L$^{-1}$ with pH of 9, an initial MB concentration of 40 mg·L$^{-1}$, and a temperature of 20 °C; (**d**) equilibrium adsorption of different supported ratios.

As indicated in Figure 6c,d the removal rate and adsorption capacity of MB increases as the mass ratio of aluminum hydroxide/iron increases, indicating that hydrotalcite could prevent nZVI from aggregating together and helps nZVI@H keep a higher adsorptivity at the same time. The removal of MB reaches 99.6% after 30 min at the mass ratio of hydrotalcite to iron of 1:1. When the mass ratio of both is increased to 2:1, the removal of MB is not significantly improved. Meanwhile, the adsorption capacity rises slightly from 79.68 mg·g$^{-1}$ to 79.76 mg·g$^{-1}$. Thus, the optimal ratio for completing subsequent experiments is determined to be 1:1.

## 4. Analysis of Factors

### 4.1. Effect of pH

As illustrated in Figure 7a, the removal rate and adsorption capacity of the nZVI@H are greatly enhanced when the pH of the MB solution is adjusted from 5 to 11, with the removal rate improving from 80.7% to 99.8% and the adsorption capacity increasing from 64.56 mg·g$^{-1}$ to 79.84 mg·g$^{-1}$. This is due to the fact that, under acidic conditions, a large number of H$^+$ competes for the adsorption sites with the cationic dyes [29]. Simultaneously, nZVI@H possesses a positive surface charge at low a pH, resulting in electrostatic repulsion between the adsorbent and dye. Conversely, the number of negative charge sites increases with the enhanced alkalinity of the solution, which promotes a strong electrostatic interaction between MB molecules and nZVI@H [30,31].

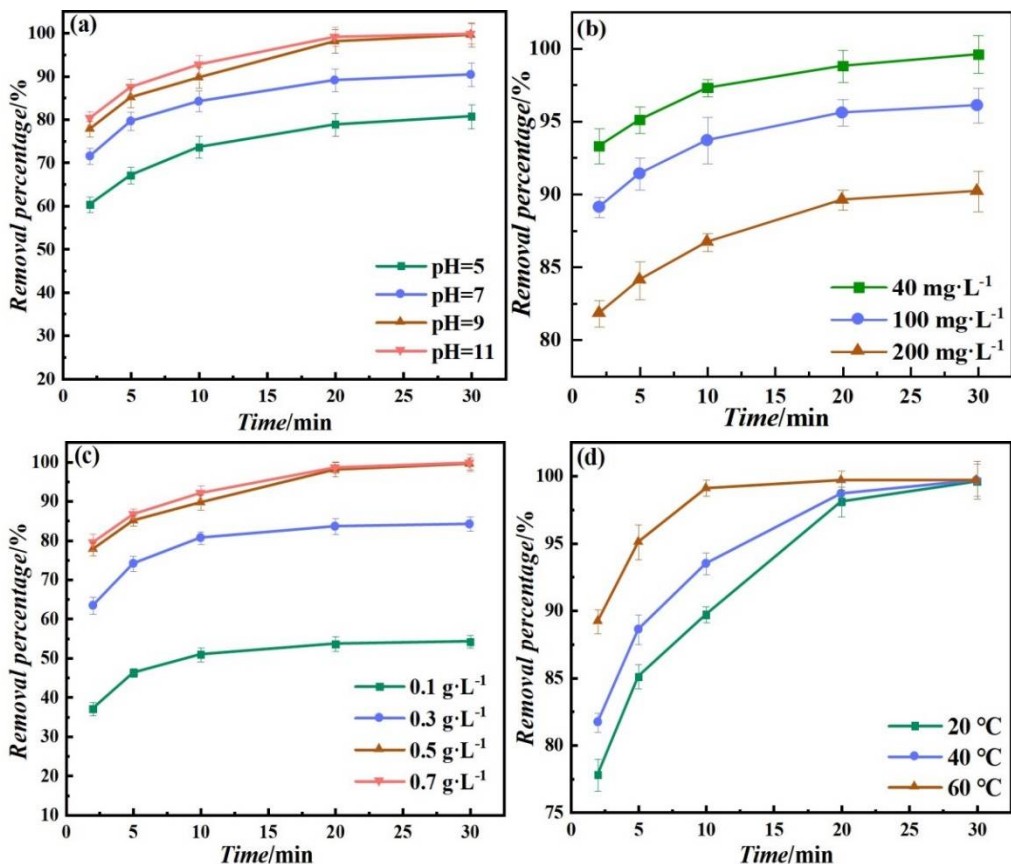

**Figure 7.** (**a**) Effect of pH on degradation of MB, in the context of 0.5 g·L$^{-1}$ nZVI@H with a temperature of 20 °C and an initial MB concentration of 40 mg·L$^{-1}$; (**b**) effect of initial concentration on degradation of MB, at the original pH (9) and temperature (20 °C) using 0.5g·L$^{-1}$ nZVI@H; (**c**) effect of dosage on degradation of MB, with a pH of 9, an initial MB concentration of 40 mg·L$^{-1}$ and a temperature of 20 °C; (**d**) effect of temperature on degradation of MB, in the context of 0.5 g·L$^{-1}$ nZVI@H with a pH of 9 and an initial MB concentration of 40 mg·L$^{-1}$.

### 4.2. Effect of Initial Concentration

Figure 7b shows the removal rates of nZVI@H for different concentrations of MB. The maximum decolorization efficiencies after 30 min are 99.6%, 96.1% and 90.2% at initial concentrations of 40 mg·L$^{-1}$, 100 mg·L$^{-1}$ and 200 mg·L$^{-1}$, respectively. It is evident that the removal rate of MB using nZVI decreases with an increase in the initial concentration of MB. This is because the adsorption point on the nZVI@H surface saturates due to the increase in the initial concentration of MB. The adsorption amounts of nZVI@H corresponding to each concentration are 79.7 mg·g$^{-1}$, 191.0 mg·g$^{-1}$ and 368.8 mg·g$^{-1}$. This behavior occurs because the increase in the initial concentration of MB enhances its driving force and thus exceeds the mass transfer resistance of MB molecules to reach the nZVI@H surface [32,33]. Therefore, as the starting concentration of MB increased, so did the adsorption capability of nZVI@H.

### 4.3. Effect of Dosage

Figure 7c shows the removal rates of MB with the nZVI@H addition of 0.1 g·L$^{-1}$, 0.3 g·L$^{-1}$, 0.5 g·L$^{-1}$ and 0.7 g·L$^{-1}$. It follows that the higher the nZVI@H dosage, the higher the removal rate of MB. After 30 min, 99.6% of MB is removed using a 0.5 g·L$^{-1}$ dosage of nZVI@H, while the removal rate is only 57.2% when the dosage of nZVI@H is decreased to 0.1 g·L$^{-1}$. This is because the adsorption process happens at the Fe$^0$–H$_2$O interface, and the adsorption surface area and the number of active sites rise as the nZVI@H dose increases. However, when the dose of nZVI@H exceeds 0.5 g·L$^{-1}$, it has little effect on the final removal rate. Therefore, 0.5 g·L$^{-1}$ is chosen as the appropriate dose combining the cost of nZVI@H and the removal rate of MB.

### 4.4. Effect of Temperature

Figure 7d shows that the final removal percentages of MB are 99.6%, 99.7%, and 99.7%, respectively. In the 20 min of the reaction, the removal percentage of MB increased with increasing temperature, and the adsorption amounts were 73.48 mg·g$^{-1}$, 75.68 mg·g$^{-1}$, and 79.76 mg·g$^{-1}$. This indicates that the adsorption capability of nZVI@H increases as temperature rises (an increase from 20 °C to 60 °C). It also confirms the endothermic character of the adsorption reaction. With temperature increases, the activation energy barrier decreases the increased rate of diffusion of MB molecules into adsorbent, thus increasing the adsorption rate and the adsorption capacity [34,35]. However, because adsorption is close to equilibrium, the final removal rate shows no significant difference between the different temperatures.

### 4.5. Isothermal Equation Analysis

The experimental data on the effect of temperature on the removal rate is used to fit the isothermal equation. The fitting results are shown in Figure 8a,b. The fitting correlation coefficient $R^2$ of the Langmuir model is 0.99 at different temperatures, as shown in Figure 8 and Table 2, while the $R^2$ of the Freundlich model is only 0.75~0.85 at different temperatures. Therefore, the Langmuir model can better describe the removal process of MB using nZVI@H. In addition, the $Q_{max}$ calculated from the Langmuir model is 79.897 mg·g$^{-1}$, 80.189 mg·g$^{-1}$, and 81.024 mg·g$^{-1}$, respectively. Furthermore, the basic properties of the Langmuir model can also be described using the dimensionless constant separation factor $R_L$ ($R_L = 1 \cdot (1 + bC_0)^{-1}$, where $C_0$ is the initial concentration of the pollutant (mg·L$^{-1}$), and b is the Langmuir adsorption constant (L·mg$^{-1}$)) [36]. According to this calculation, the $R_L$ at three temperatures is 0.009, 0.004, and 0.001, respectively. All three $R_L$ values ranged from 0 to 1, indicating that removing MB by nZVI is a reaction process that favors adsorption.

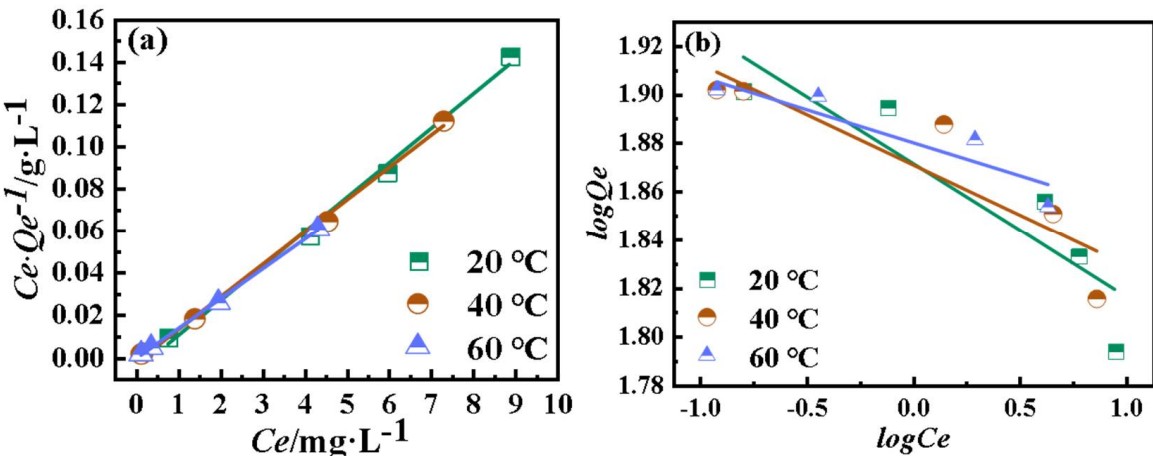

**Figure 8.** Linear fitting of (**a**) Langmuir model and (**b**) Freundlich model.

**Table 2.** Fitting results of isothermal equation at different temperatures.

| | Langmuir | | | Freundlich | | |
|---|---|---|---|---|---|---|
| $T$ (°C) | $Q_{max}$ (mg·g$^{-1}$) | $B$ (L·mg$^{-1}$) | $R^2$ | $K_f$ ((mg·g$^{-1}$)/(L·mg$^{-1}$)$^{1/n}$) | $n^{-1}$ (mg·L$^{-1}$) | $R^2$ |
| 20 | 79.897 | 2.638 | 0.9914 | 73.394 | 0.0552 | 0.7543 |
| 40 | 80.189 | 6.313 | 0.9960 | 74.290 | 0.0416 | 0.7607 |
| 60 | 81.024 | 20.985 | 0.9988 | 75.872 | 0.0273 | 0.8145 |

### 4.6. Adsorption Kinetic Analysis

The kinetic fitting results are shown in Figure 9, and the kinetic parameters for different initial MB concentrations (Figure 9a,b) are summarized in Table 3. As shown in Table 3, compared to that of the quasi-first-order kinetics model, the correlation coefficient ($R^2$) of the quasi-second-order kinetics model ranged from 0.99 to 1, which indicates that the adsorption process is mainly chemisorption, and the chemisorption process of its surface active site and MB is the main limiting factor of the adsorption rate. The equilibrium adsorption amounts fitted by the quasi-second-kinetics model are closer to the experimental values. The rate constants $k_2$ of the quasi-second-order kinetic equation are equal to 0.097 g·mg$^{-1}$·min$^{-1}$ for 40 mg·L$^{-1}$, 0.035 g·mg$^{-1}$·min$^{-1}$ for 100 mg·L$^{-1}$ and 0.019 g·mg$^{-1}$·min$^{-1}$ for 200 mg·L$^{-1}$, respectively. The increase in MB concentration slows down the adsorption rate, demonstrating that MB's adsorption occurs mainly at the interface with nZVI@H.

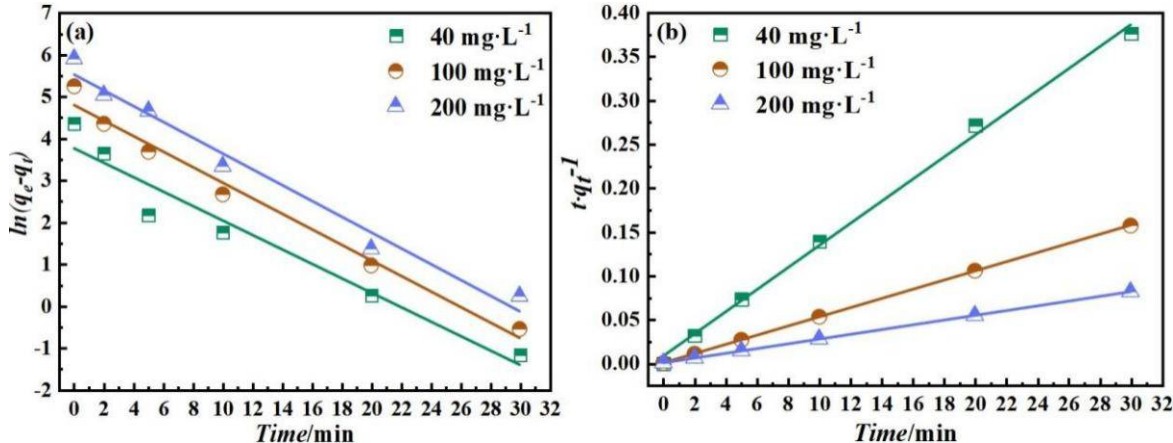

**Figure 9.** Quasi-first-order kinetics model (**a**) and quasi-second-order kinetics model (**b**).

**Table 3.** Fitting results of kinetic equation under different concentrations.

| $C_{MB}$ (mg·L$^{-1}$) | $Q_{e,exp}$ (mg·g$^{-1}$) | Quasi First-Order Kinetics Equation | | | Quasi Second-Order Kinetics Equation | | |
|---|---|---|---|---|---|---|---|
| | | $Q_{e,cal}$ (mg·g$^{-1}$) | $k_1$ (min$^{-1}$) | $R^2$ | $Q_{e,cal}$ (mg·g$^{-1}$) | $k_2$ (g·mg$^{-1}$·min$^{-1}$) | $R^2$ |
| 40 | 79.7 | 43.1 | 0.172 | 0.937 | 79.4 | 0.097 | 0.986 |
| 100 | 191.0 | 121.1 | 0.185 | 0.959 | 190.8 | 0.035 | 0.988 |
| 200 | 368.8 | 250.9 | 0.188 | 0.962 | 369.1 | 0.019 | 0.991 |

As can be seen from Figure 10, the adsorption process of MB by nZVI@H can be divided into the following three steps: (1) diffusion of MB molecules through the boundary layer across the adsorbent surface; (2) diffusion of MB molecules within the adsorbent particles by liquid filling; and (3) MB molecules reaching adsorption equilibrium. The first and second steps are generally considered extremely fast, as seen from the slopes in Table 4. Therefore, the most significant effects on the adsorption rate are boundary layer diffusion and intraparticle diffusion. The three linear stages indicate that a single diffusion mode does not control the particle diffusion rate. According to the intercept values, the absorption process of MB proceeds from diffusion across the boundary layer to intraparticle diffusion.

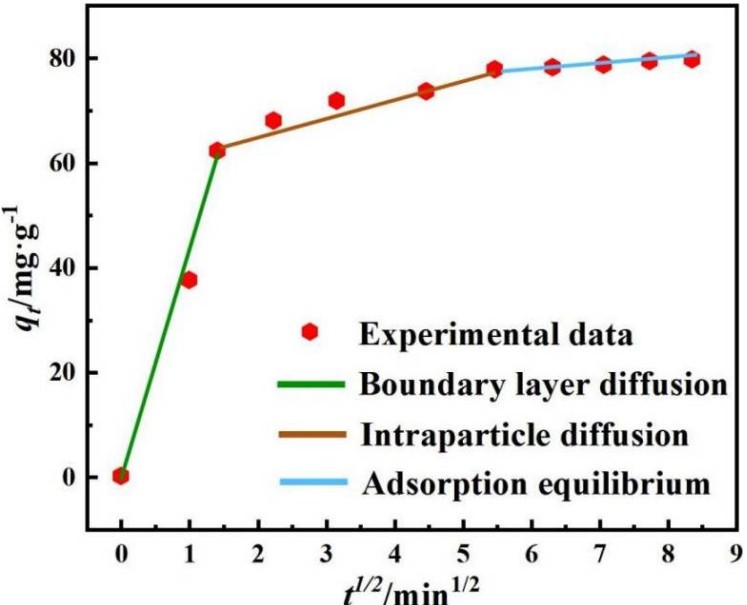

**Figure 10.** Diffusion model.

**Table 4.** Diffusion model parameters.

| Diffusion Model | Main Parameters | | |
|---|---|---|---|
| | $k_d$ (mg·(g·min$^{1/2}$)$^{-1}$) | $C$ (mg·g$^{-1}$) | $R^2$ |
| Boundary layer diffusion | 42.79 | 1.20 | 0.973 |
| Intraparticle diffusion | 3.48 | 59.01 | 0.917 |
| Adsorption equilibrium | 0.69 | 73.92 | 0.976 |

Table 5 shows the adsorbent adsorption amounts to MB from previous studies. After comparison, nZVI@H has a strong adsorption capacity.

**Table 5.** Comparison of adsorption capacity of MB on diversified adsorbents.

| Adsorbent | $Q_{max}$ (mg·g$^{-1}$) | Reference |
|---|---|---|
| Fly ash | 5.57 | [37] |
| Biochar-CNT | 6.20 | [38] |
| MWCNT-SH | 10.00 | [39] |
| Lignin and chitosan | 36.00 | [40] |
| PFB-mZVI adsorbent | 42.80 | [37] |
| Straw BC | 62.50 | [41] |
| nZVI@H | 81.02 | This study |

### 4.7. Reusability and Stability of nZVI@H

Figure 11a shows the results of magnetic detection of bare nZVI and nZVI@H using VSM. From (a), it can be seen that the saturation magnetization intensity of nZVI@H formed through the combination of magnetic nZVI with non-magnetic hydrotalcite decreased by nearly half (from 72.50 emu·g$^{-1}$ to 39.36 emu·g$^{-1}$). This confirms the SEM results that hydrotalcite contributes to the reduction of the magnetic properties of nZVI and the reduction of nZVI agglomeration. In addition, the coercivity magnetization of nZVI@H is above 20 G, which proves that it is a soft ferromagnetic material. Figure 11b shows the results of the repeated use experiment (6 cycles) of nZVI with nZVI@H. At the initial use, the removal rates of MB by nZVI and nZVI@H are 71.2% and 99.6%, respectively. Apparently, after three cycles, the removal rate of nZVI decreases to 10.1%, while nZVI@H can maintain a 93.9% rate. After six cycles, nZVI@H still has more than half of the removal rate, and this result suggests that nZVI@H has excellent stability and reusability. Furthermore, the material remains magnetic following cycles (nZVI@H-MB in Figure 11a), creating conditions for subsequent nZVI@H recycling and reuse.

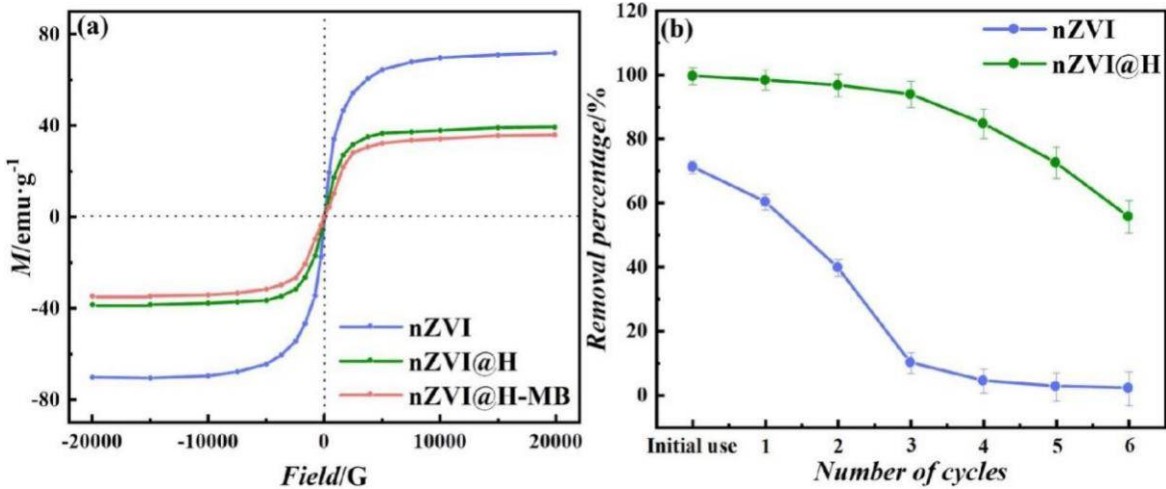

**Figure 11.** (**a**) VSM of nZVI, nZVI@H and nZVI@H-MB; (**b**) reusability of nZVI@H in MB removal.

### 4.8. MB Removal Mechanisms

Removal of MB from the solution is closely related to redox reactions, chemical adsorption and precipitation processes [28]. The removal process of MB involves the oxidation of Fe$^0$ (Equation (8)) followed by the chemical reduction of MB [42]. MB obtains electrons emitted from Fe$^0$ and turns itself into colorless leuco-methylene blue (LMB) (Figure 12). This is consistent with the kinetic analysis. As shown in Figure 12, the bond energies of ①~⑤ are 615, 536, 389, 305, and 272 kJ·mol$^{-1}$, respectively. The C=N and C=S with high-bond energies are broken into C–N and C–S with low-bond energies by the release of electrons from the Fe$^0$. At the same time, the N in the MB molecule forms a bond with the H$^+$ in the water (N–H) to produce the LMB. It is the main reason for the decolorization

of the dye. Then a series of small molecular substances are generated and separated using adsorption coprecipitation through the following reactions (Equations (10)–(13)):

$$Fe^0 \rightarrow Fe^{2+} + 2e^- \tag{8}$$

$$nZVI + MB \text{ and } LMB \rightarrow nZVI\text{-}(MB \text{ and } LMB) \tag{9}$$

$$nZVI@H + MB \text{ and } LMB \rightarrow nZVI@H(MB \text{ and } LMB) \tag{10}$$

$$Fe^{2+} + MB \text{ and } LMB \rightarrow Fe(II)(MB \text{ and } LMB) \tag{11}$$

$$Fe^{3+} + MB \text{ and } LMB \rightarrow Fe(III) \ (MB \text{ and } LMB) \tag{12}$$

$$nFe_x(OH)_y^{(3x-y)} + MB \text{ and } LMB \rightarrow (MB \text{ and } LMB)[Fe_x(OH)_y^{(3x-y)}]_n \tag{13}$$

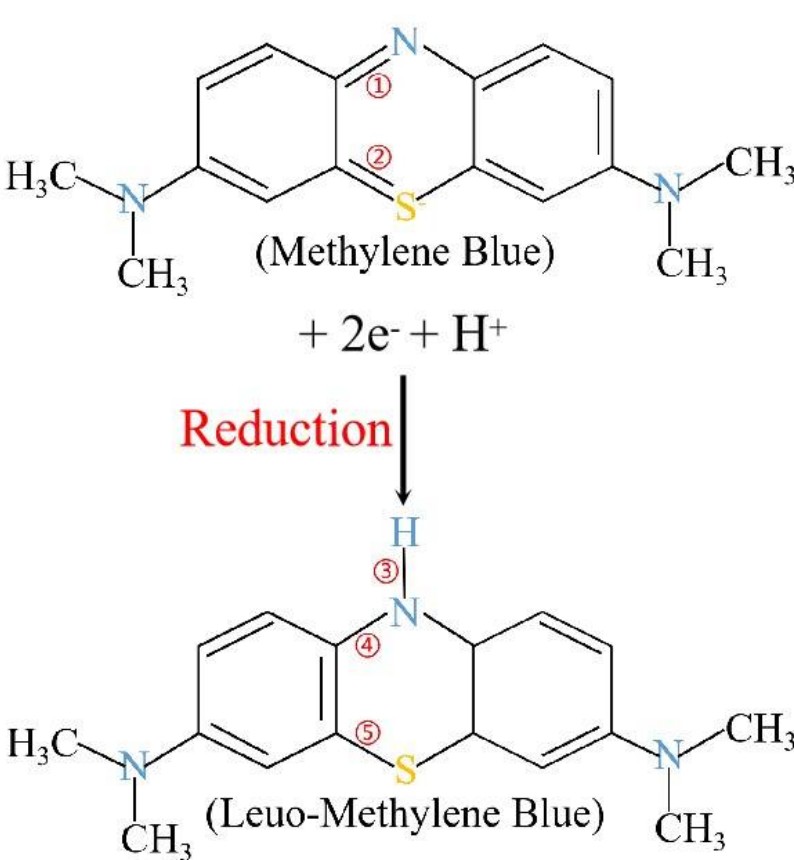

**Figure 12.** Decolorization reaction of MB.

### 5. Conclusions

In this study, hydrotalcite is treated as a potential supporter, and nZVI@H is successfully prepared using the liquid-phase reduction process method. Using nZVI@H, the removal mechanism of MB has been insightfully investigated, in conjunction with a series of characterized methods, such as SEM, BET, XRD, FTIR, zeta-potential analyses and VSM. The results showed that hydrotalcite contributes to the dispersion characteristic of nZVI particles, which is in favor of the removal of MB. Compared to the bare nZVI, nZVI@H shows the higher removal of MB. After 30 min of reaction, the removal rate of MB can reach 99.6% at an initial MB concentration of 40 mg·L$^{-1}$, while the removal rate of bare nZVI is only 71.2%. According to this mechanistic study, the adsorption process of nZVI@H on MB matches well with the Langmuir and quasi-second-order kinetics models. The maximum adsorption capacity of nZVI@H is calculated to be 81 mg·g$^{-1}$ based on the Langmuir model, which is highly superior to most reported adsorbent materials. Simultaneously, MB can be converted to colorless LMB through the Fe$^0$ redox reaction process, and finally precipitated

with nZVI@H. Therefore, the removal process of MB by nZVI@H can be involved in three processes: a redox reaction, chemical adsorption and complexation precipitation. In addition, reusability experiments confirm the good reusability of nZVI@H, and it remains magnetic after six cycles, which facilitates subsequent recycling.

**Author Contributions:** J.F. and B.Z. (Bo Zhang) contributed to the conception of the study; J.F. collected data and wrote the initial paper; B.Z. (Bohong Zhu), W.S., Y.C. and F.Z. revised the paper. All authors have read and agreed to the published version of the manuscript.

**Funding:** This work was supported by the National Special Foundation for Sustainable Development Agenda Innovation Demonstration Zone Construction, Technology Development and Demonstration of Nanoscale Zero-Valent Iron Preparation and Water Remediation (2019sfq27), Graduate Innovation Project of Hunan University of Technology (CX2180), the Hunan Natural Science Foundation of China (No. 2022JJ40143), and Innovative entrepreneurial projects for college students (202111535003S).

**Institutional Review Board Statement:** Not applicable.

**Informed Consent Statement:** Not applicable.

**Data Availability Statement:** This article contains all of the data generated or analyzed during this study.

**Acknowledgments:** The authors would like to thank the National Special Foundation, the Hunan Natural Science Foundation of China, the innovative entrepreneurial projects for college students and Hunan University of Technology for supporting of this work.

**Conflicts of Interest:** The authors declare that they do not have any competing interests.

**Ethics Approval:** We confirm that we saw and approved the submitted manuscript. Our manuscript does not include or report on any animal or human data or tissue.

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
