# Peer review of "New Insight into the Mechanism for the Removal of Methylene Blue by Hydrotalcite-Supported Nanoscale Zero-Valent Iron"

_water, doi:10.3390/w15010183_

Round 1

Reviewer 2 Report

This manuscript focus on the methylene blue removal by hydrotalcite supported nanoscale zero-valent iron, which is of interesting to broad readers. After carefully reading, it was found that this article needs major revisions because several issues and explanations are still need to be clarified.

1.      “MB remediation” in line 18 is suggested to be replaced by “MB removal”.

2.      Why does hydrotalcite supported nZVI abbreviated as Nzvi@S? What is the relation between “S” and hydrotalcite? It is kind of weird.

3.      Waste water treatment is important for the sustainable development. Various absorbents have been developed for dye removal, heavy metal ions removal, etc. More references are suggested to be cited for broad readers, e.g. Journal of Bioresources and Bioproducts 2021, 6 (4), 292-322; Journal of Bioresources and Bioproducts 2022, 7 (2), 109-115; Chemical Engineering Journal 2022, 446, 136851.

4.      “The main reaction conditions involved in this experiment wae as follows” in line 110-111 needs to be revised.

5.      A space should be added between the number and the unit for “2min, 5min, 10min, 20min and 30min” in line 128.

6.      A full stop should be added for the titles of figures and tables.

7.      The font sizes of “Different materials”, “Supported Ratio” and the numbers in (b) and (d) in Figure 6 are suggested to be the same as “Time/min”.

8.      “mg/L” in Figure 7(b) should be revised as “mg·L-1”.

9.      “kJ/mol” in line 317 should be revised as “kJ·mol-1”. Please check the whole manuscript to fix the same issues.

10.   Does nZVI@S could be recycled and reused? How about the performance in different cycle?

11.   Please pay attention to the writing of chemical formula in references.

        12. Please double check the references. Page numbers are missing for many references. “Huan jing ke xue= Huanjing kexue” in ref.14 need to be revised.

Round 2

Reviewer 1 Report

The article was improved and can be accepted after minor revision

1- The Qmax in abstract and Table 5 must be related to the Langmuir model for  real comparison with different adsorbents

2- The KF unit of the Freundlich model in Table 2 must be corrected

Reviewer 2 Report

The manuscript is acceptable by fixing the reference  [2], [4], and [6]. Please write them in the same way as other references.
